# Associations between lean maturity in primary care and musculoskeletal complaints among staff: a longitudinal study

Monica Kaltenbrunner [1,2] Svend Erik Mathiassen,[1] Lars Bengtsson,[3] Hans Högberg,[1] Maria Engström[1,4]

¹Faculty of Health and Occupational Studies, University of Gävle, Gavle, Sweden
²Faculty of Health and Life Sciences, Linnaeus University, Kalmar, Sweden
³Faculty of Engineering and Sustainable Development, University of Gävle, Gavle, Sweden
⁴Department of Public Health and Caring Sciences, Uppsala University, Uppsala, Sweden

**Correspondence to**
Dr Monica Kaltenbrunner;
monica.kaltenbrunner@hig.se

## ABSTRACT

**Objective** This study had two aims: (1) to determine the prevalence of musculoskeletal complaints among staff in primary care and (2) to determine to what extent lean maturity of the primary care unit can predict musculoskeletal complaints 1 year later.

**Design** Descriptive, correlational and longitudinal design.

**Setting** Primary care units in mid-Sweden.

**Participants** In 2015, staff members responded to a web survey addressing lean maturity and musculoskeletal complaints. The survey was completed by 481 staff members (response rate 46%) at 48 units; 260 staff members at 46 units also completed the survey in 2016.

**Outcome measures** Associations with musculoskeletal complaints were determined both for lean maturity in total and for four Lean domains entered separately in a multivariate model, that is, philosophy, processes, people and partners, and problem solving.

**Results** The shoulders (12-month prevalence: 58%), neck (54%) and low back (50%) were the most common sites of 12-month retrospective musculoskeletal complaints at baseline. Shoulders, neck and low back also showed the most complaints for the preceding 7 days (37%, 33% and 25%, respectively). The prevalence of complaints was similar at the 1-year follow-up. Total lean maturity in 2015 was not associated with musculoskeletal complaints, neither cross-sectionally nor 1 year later, for shoulders (1 year β: −0.002, 95% CI −0.03 to 0.02), neck (β: 0.006, 95% CI −0.01 to 0.03), low back (β: 0.004, 95% CI −0.02 to 0.03) and upper back (β: 0.002, 95% CI −0.02 to 0.02).

**Conclusion** The prevalence of musculoskeletal complaints among primary care staff was high and did not change within a year. The extent of lean maturity at the care unit was not associated with complaints among staff, neither in cross-sectional analyses nor in a 1-year predictive analysis.

## INTRODUCTION

Implementation of lean has become common in healthcare,[1] with an aim to increase efficiency[2] and quality of care.[3 4] Lean originates from the automotive industry[5] and has been described and construed in different ways, both there and in healthcare.[1 4 6 7]

## STRENGTHS AND LIMITATIONS OF THIS STUDY

⇒ The longitudinal design allowed analysis over time of associations between lean maturity and self-reported musculoskeletal complaints.

⇒ The convenience sample and the low response rate are potential limitations.

⇒ Use of self-reports may be a limitation, as responses may be biased by a positive or negative attitude towards lean and the workplace.

Several systems for describing lean have been suggested. Liker's interpretation, which we use in the present study,[5] includes 14 basic principles, which can be organised into four categories in a so-called 4P model (philosophy, processes, people and partners, and problem solving). 'Philosophy' involves having a long-term perspective, focusing on the customer, and having common goals for the staff members to strive for in order to create value for the customer, the organisation and society as a whole. 'Processes' involve using resources optimally to improve flow in production. A good flow obtained by, for instance, mapping processes, standardising work and reducing waste can increase quality and efficiency. 'People and partners' include showing respect for individuals in the organisation and enabling them to grow, as well as respecting suppliers and partners. 'Problem-solving' focuses on finding and solving basic causes of problems restraining flow and quality in production.[5]

In lean, staff members are given responsibility, and they are expected to contribute to decision making and to engage in securing continuous improvements. Customer satisfaction is paramount, and according to Liker, all staff throughout the organisation need to be aware of that and contribute to achieving it.[5] Liker also claims that all lean principles must

be adopted throughout the organisation in order to fully reach the desired results.[5] Thus, the ultimate level of lean maturity will be obtained when all staff adopts all lean principles to their full extent.[8]

In Sweden, lean was first implemented in industry and began influencing healthcare management in the mid-90s.[9] However, several studies in healthcare indicate that lean has only been partially adopted, with an emphasis on, for example, standardised work, enhanced seamlessness and flow, and problem solving.[1 4 10 11] Some examples have been reported of positive effects of adopting lean, including that lean may improve working conditions[12] and patient safety.[13 14] In a previous study in primary care, we showed that lean maturity was positively associated with quality of care as perceived by staff.[8] However, other studies conclude that the effects of implementing lean are ambiguous.[15 16] In the context of work-related complaints, lean should, according to Liker,[5] to address human resources in the sense that lean should not contribute to excessive loads on the staff, leading to negative health consequences, including, for example, musculoskeletal complaints.[17] However, two reviews[17 18] and one overview[19] suggest that Lean may, indeed, contribute to increased musculoskeletal complaints, at least in other sectors than healthcare. This might be a result of lean influencing working conditions so that work demands and work pace increase. This, in turn, may lead to decreased opportunities for recovery, which may increase the risk for musculoskeletal complaints.[17]

Little is known, however, about associations between Lean and musculoskeletal complaints specifically in healthcare.[1 2 10] Musculoskeletal complaints among healthcare staff have been reported to be common[20] and a concern.[21] For instance, a review of musculoskeletal complaints among registered nurses, midwives and physicians working predominantly in hospitals reported that 45% had musculoskeletal complaints in the neck, 40% in the shoulders and 35% in the upper back during a 12-month period, as assessed using the Nordic Musculoskeletal Questionnaire (NMQ).[22] In a cross-sectional study by Freimann et al[23] among Finnish nurses, the 12-month prevalence of musculoskeletal complaints based on the NMQ was 57% for the low back, 56% for the neck and 31% for the shoulders.

In healthcare, likely risk factors for musculoskeletal complaints among staff working in hospitals and elderly care are awkward postures, repetitive work, heavy lifting, role conflicts and high emotional demands together with low influence, lack of respect and lack of justice.[20 23–29] In turn, these risks can be modified by gender, age and body mass index (BMI).[30] Primary care, however, represents working conditions differing from those in hospitals. In Sweden, primary care offers prevention, advice and treatment to the citizens, and it is the first level of care in the healthcare system.[25] Working in primary care has been described as having low physical demands, while cognitive demands are substantial.[31] Thus, psychosocial working conditions such as low decision authority, low social support, high job demands, high job strain and psychological distress—all risk factors for musculoskeletal disorders[32]—may prevail in primary care but may be expressed different than in hospitals. Differences in physical and psychosocial risk factors between primary care and other healthcare areas may lead to differences in complaints and thus different corrective actions.

Few studies have, however, investigated this issue. A cross-sectional study in Portuguese primary care, based on the NMQ, reported that 63% of the participating nurses had low back pain, 50% neck pain and 41% pain in the dorsal region during the preceding 12-month period.[26] In Swedish primary care, the prevalence of musculoskeletal complaints has, to our knowledge, not been assessed in any scientific study. Since lean sets the scene for working conditions among staff, the association between lean maturity and musculoskeletal complaints in primary care stands up as an interesting issue for research. Hence, the aim of the present study was twofold: (1) to determine the prevalence of musculoskeletal complaints among staff in primary care and (2) to determine to what extent lean maturity of the primary care unit could predict musculoskeletal complaints 1 year later.

## METHODS

### Primary care units

The study had a descriptive, correlational and longitudinal design. A convenience sample of 52 primary care units located in a region in mid-Sweden were first approached; 42 units accepted, of which 38 were public primary care units and four belonged to private providers. Due to this shortage of private care units, one of the largest nationwide private healthcare providers (organising 85 primary care units) was approached; six units from there agreed to participate, in addition to the four already recruited. Units were required to, according to their own opinion, have adopted lean to some extent to be eligible for participation in the study.

### Participants

All staff at the recruited units were asked to participate and responded to a questionnaire in May 2015 (T1) and again in May 2016 (T2), with an inclusion criterion of having worked at the unit at least 3 months prior to data collection. The number of staff eligible at T1 was 1040; 481 responded, distributed among 48 units, that is, a 46% response rate. At T2, 406 staff members of those who responded at T1 were still eligible. In total, 260 staff members at 46 units responded at both T1 and T2 (figure 1).

### Data collection procedure

Data were collected at T1 and T2 using a web survey. After two non-successful reminders, non-responders were sent a paper version. Confidentiality and voluntary participation were emphasised at data collection. Consent to participate was given when participants completed and

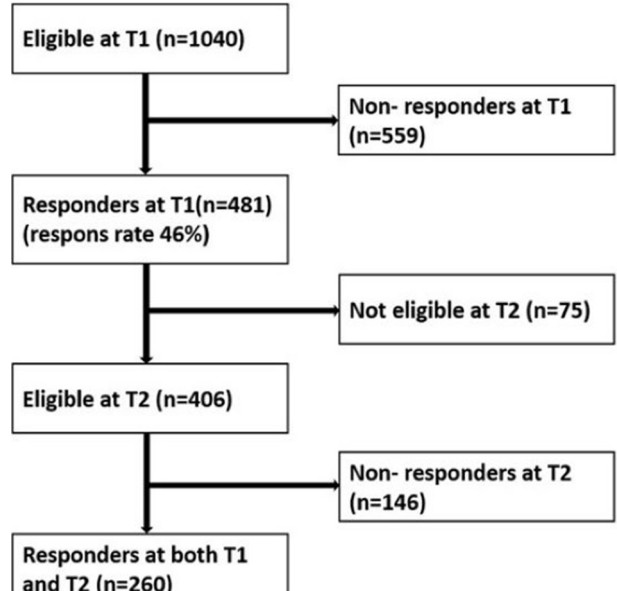

**Figure 1** Flow chart showing staff members eligible for the present study, and the flow of responding staff members between T1 and T2. T1, baseline (2015); T2, follow-up (2016).

returned the questionnaire. The study is part of a larger research project on lean maturity in primary care; data have been used and published for other purposes in previous papers.[8 33 34]

## Measurements

Staff members' self-rated perception of *lean maturity* at their unit was assessed using the 16-item Lean in Health care Questionnaire (LiHcQ), shown to have acceptable psychometric properties.[33] The LiHcQ is based on the 4P model of Lean proposed by Liker,[5] addressing philosophy (three items), processes (seven items), people and partners (three items) and problem solving (three items). Answers are provided on a scale ranging from 1, indicating low lean maturity, to 5, indicating high lean maturity. Thus, the total LiHcQ score could range from 16 to 80. In the present study, Cronbach's alpha was 0.93 for total lean maturity, and for philosophy 0.75, processes 0.86, people and partners 0.60 and problem solving 0.81, respectively. *Musculoskeletal complaints* (pain and/or discomfort) were assessed using the NMQ.[24] All 18 items in the first part of the NMQ were included; they address, with response alternatives 'yes' and 'no', the occurrence of musculoskeletal complaints during the past 12-month and 7-day periods in nine body regions (neck, shoulders, low back, upper back, elbows, hands/wrists, hips, knees and ankles/feet). In the second part of the NMQ (which in the present study addressed pain only in the neck, shoulders and low back), the respondent is asked to respond with a 'yes' or 'no' to whether she, during the preceding year, has been limited in performing activities during work or leisure, has changed jobs or work tasks or has visited a health practitioner due to musculoskeletal complaints. Items also address the total duration of musculoskeletal complaints during the preceding year

(five-point scale, with a 1 for 0 days and a five for every day). Participants were also asked to report gender, age, height, weight, years worked in the profession and years worked at the present primary care unit. Lean maturity as well as musculoskeletal complaints were assessed both at T1 and T2.

## Statistical analysis

Data analyses were conducted using IBM SPSS Statistics, V.24. Descriptive statistics were used for demographic data, the NMQ and LiHcQ. Differences between T1 and T2 in musculoskeletal complaints were analysed using McNemar's test; changes in lean maturity over time were analysed using a paired t-test. Associations between variables in the cross-sectional as well as the longitudinal case were determined using generalised estimating equations, with an exchangeable correlation structure accounting for the correlated nature of data collected at the same primary care unit. Predictive analyses over time were conducted using LiHcQ scores at T1 as the predictor variable, and musculoskeletal complaints at T2 as the outcome, controlling for complaints at T1. Additional control variables were gender, BMI and age categorised in quartiles. Since musculoskeletal complaints showed to be most prevalent in shoulders, neck, low back and upper back, these body regions were selected for analysis of associations between lean maturity and complaints. Two models were tested based on cross-sectional data and four models based on longitudinal data (table 1).

Missing data were handled using multiple imputation (MI).[35] When conducting a MI, five new datasets are produced with imputed data. In the subsequent analysis, results from the original dataset and the five MI datasets are generated, as well as a pooled result. We report the pooled results, in terms of regression coefficients with 95% CIs and corresponding p values. SD and ORs were obtained on basis of the five MI datasets, since pooled results do not provide these metrics.

## Patient and public involvement

None.

## RESULTS

### Participants

The sample at T1 consisted mostly of females; their mean age was 50.2 years (SD 10.3) (table 2). The most common profession was registered nurse, and most of the participants worked in public primary care units. Non-responders at T2, that is, those only responding at T1, did not differ significantly from those responding at both T1 and T2 with respect to gender, age, years worked in the profession and years worked at the present unit.

### Prevalence of musculoskeletal complaints

At T1, 50%–58% of the participants (n=481) reported musculoskeletal complaints in the preceding 12-month period in the shoulder, neck or low back (table 3).

**Table 1** Variables included in the six models addressing associations between lean maturity and musculoskeletal complaints

| | Cross-sectional data | | Longitudinal data | | | |
|---|---|---|---|---|---|---|
| | Model 1a | Model 1b | Model 2a | Model 2b | Model 2c | Model 2d |
| **Independent variables** | | | | | | |
| Lean maturity at T1, in total | × | | × | × | | |
| Lean maturity at T1, each of the 4P | | × | | | × | × |
| **Dependent variables** | | | | | | |
| Complaints at T1 | × | × | | × | | × |
| Complaints at T2 | | | × | × | × | × |

All six models included gender, body mass index and age (in quartiles) as control variables.
4P, philosophy, processes, people and partners, and problem solving; T1, time 1 (2015); T2, time 2 (2016).

**Table 2** Participant characteristics

| | T1 N: 481 | (%) | T2 N: 260 | (%) |
|---|---|---|---|---|
| **Participants working at** | | | | |
| Public primary care units | 407 | (85) | 216 | (83) |
| Private primary care units | 74 | (15) | 44 | (17) |
| **Gender** | | | | |
| Women | 422 | (88) | 224 | (86) |
| Men | 59 | (12) | 35 | (13) |
| **Profession** | | | | |
| Registered nurses | 181 | (38) | 101 | (39) |
| Physician | 70 | (15) | 37 | (14) |
| Administrator and secretary | 64 | (13) | 31 | (12) |
| Physiotherapist | 47 | (10) | 30 | (12) |
| Licensed practical nurse | 45 | (9) | 24 | (9) |
| Social worker and psychologist | 41 | (9) | 21 | (8) |
| Manager | 26 | (5) | 17 | (7) |
| Occupational therapist | 17 | (4) | 8 | (3) |
| Dietitian | 3 | (1) | 1 | (< 1) |
| **Age** | | | | |
| Mean (SD) | 50.2 (10.3) | | 50.6 (10.0) | |
| Md ($Q_1$–$Q_3$) | 52.0 (44.0–59.0) | | 53.0 (44.0–59.0) | |
| **BMI** | | | | |
| Mean (SD) | 25.0 (4.0) | | 25.4 (4.1) | |
| Md ($Q_1$–$Q_3$) | 24.2 (22.4–27.4) | | 24.6 (22.6–27.8) | |
| **Years worked in the profession** | | | | |
| Mean (SD) | 21.5 (12.1) | | 22.0 (11.8) | |
| Md ($Q_1$–$Q_3$) | 21.0 (11.0–31.0) | | 20.0 (13.0–32.0) | |
| **Years worked at the present unit** | | | | |
| Mean (SD) | 9.1 (9.0) | | 9.3 (8.7) | |
| Md ($Q_1$-$Q_3$) | 5.0 (2.0–14.0) | | 5.0 (3.0–14.0) | |

Numbers regarding professions do not add up to 481 and 260 because some participants had multiple professions.
BMI, body mass index; Md, median; T1, time 1 (2015); T2, time 2 (2016).

**Table 3** Prevalence of musculoskeletal complaints at T1 (n=481)

| | Musculo-skeletal complaints | Musculo-skeletal complaints during the preceding 7-day period | Musculoskeletal complaints limiting activities during…* | | Changed job or work tasks due to musculoskeletal complaints* | Visited a health practitioner due to musculoskeletal complaints* |
| | | | leisure | work | | |
| | n (%) | n (%) | n (%) | n (%) | n (%) | n (%) |
|---|---|---|---|---|---|---|
| Shoulders | 264 (58) | 176 (37) | 86 (18) | 30 (6) | 23 (5) | 86 (18) |
| Neck | 249 (54) | 157 (33) | 77 (16) | 35 (7) | 15 (3) | 75 (16) |
| Low back | 222 (50) | 119 (25) | 102 (21) | 44 (9) | 33 (7) | 70 (15) |
| Upper back | 160 (36) | 74 (15) | | | | |
| Wrists/hands | 133 (30) | 81 (17) | | | | |
| One hip or both hips | 121 (27) | 72 (15) | | | | |
| One knee or both knees | 121 (27) | 71 (15) | | | | |
| One or both ankles/feet | 112 (25) | 76 (16) | | | | |
| Elbows | 54 (12) | 29 (6) | | | | |

Data concern the preceding 12-month period if not stated otherwise.
*These outcomes were obtained only for shoulders, neck and low back, as explained in the running text. All percentages refer to n=481.

Musculoskeletal complaints were also most common in these three body regions when reported for the preceding 7 days (table 4). The same pattern was found when analysing only those who responded at both T1 and T2 (n=260).

At T1 (n=481), musculoskeletal complaints during the preceding 12-month period most often lasted for 1–7 days for the low back (21%) and neck (20%), while shoulder complaints most often lasted for more than 30 days (15%). Nine per cent reported daily musculoskeletal complaints in the past 12-month period in the shoulders, 7% in the low back and 6% in the neck.

### Changes in musculoskeletal complaints
Changes over time in musculoskeletal complaints were analysed for shoulders, neck, low back and upper back. Between 20% and 43% of the participants reported musculoskeletal complaints from these body regions at both T1 and T2 (table 4). Between 12% and 13% reported musculoskeletal complaints in the four body regions at T1, which were not present anymore at T2. Between 11% and 14% had no musculoskeletal complaints at T1 but reported complaints at T2. Most of the participants (72%–76%) did not change their ratings between T1 and T2. Differences between T1 and T2 in musculoskeletal complaints were non-significant for all four body regions (table 4).

### Lean maturity
Mean lean maturity was 46.2 at T1 (SD ranging from 11.5 to 11.7 in the MI dataset) and 45.3 at T2 (SD ranging from 11.9 to 12.1). Between T1 and T2, 94.5% of all participants changed their rating of total lean maturity; however, these changes were both positive and negative (figure 2). Thus, no statistically significant difference in

**Table 4** Combined prevalence of musculoskeletal complaints at T1 and T2, including results of McNemar's test for differences in ratings between T1 and T2

| | Shoulders (n=260) | Neck (n=260) | Low back (n=260) | Upper back (n=260) |
|---|---|---|---|---|
| Musculoskeletal complaints at T1 and T2 | 110 (42%) | 113 (43%) | 94 (36%) | 51 (20%) |
| Musculoskeletal complaints at T1 but not at T2 | 33 (13%) | 31 (12%) | 34 (13%) | 32 (12%) |
| No musculoskeletal complaints at T1 but complaints at T2 | 37 (14%) | 32 (12%) | 37 (14%) | 29 (11%) |
| Musculoskeletal complaints neither at T1 nor at T2 | 80 (31%) | 84 (32%) | 94 (36%) | 147 (57%) |
| McNemar test, original dataset, p value | 0.892 | 0.609 | 0.775 | 0.788 |
| MI-dataset 1–5, p range | 0.899 to 1.000 | 0.470 to 0.906 | 0.366 to 1.000 | 0.550 to 1.000 |

P calculated with binomial distribution.
MI, multiple imputation; T1, time 1; T2, time 2.

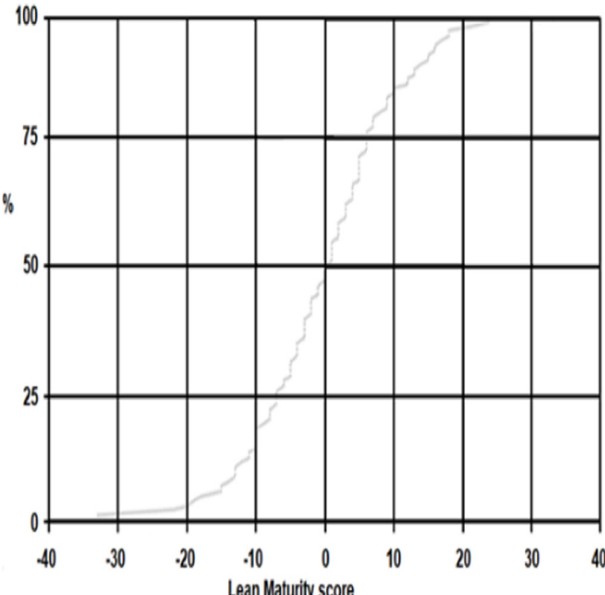

**Figure 2** Changes in individual ratings of lean maturity between T1 and T2 for non-imputed data (n=104). The value 0 indicates no change in rating between T1 and T2. Negative and positive values show a decrease and an increase, respectively, in lean maturity from T1 to T2.

lean maturity was found at the group level between T1 and T2 (pooled data; p=0.16).

### Associations between lean maturity and musculoskeletal complaints

Cross-sectional data at T1 showed no statistically significant associations between lean maturity in total and musculoskeletal complaints in the shoulders, neck, low back or upper back (p between 0.09 and 0.96; table 5, model 1a). Using each of the 4P as the independent variable instead of lean maturity in total did not result in any statistically significant associations either, except for neck complaints showing a positive association with philosophy (p≤0.01; table 5, model 1b) and a negative association with people and partners (p=0.03).

Similarly, when analysing longitudinal data, we found no statistically significant associations between lean maturity in total at T1 and musculoskeletal complaints in the four body regions at T2 (p from 0.57 to 0.88) (tables 6–9). When analysing associations between complaints at T2 and each of the individual 4P at T1, adding complaints at T1 as a covariate, people and partners were positively associated with neck complaints (p=0.03) (table 7, model 2d).

### DISCUSSION

The 12-month prevalence of low back pain in hospital settings has been reported to be higher than what we found in the present study,[22][23][36][37] whereas the prevalence of shoulder complaints appears lower in hospitals.[22][23] In a study of different occupations in Iran, the prevalence of low back pain among healthcare staff was 58%,[37] that is,

larger than in our sample. Only one study was conducted in a Swedish setting.[36] A study by Ribeiro et al[26] from primary care in Portugal reported a 63% prevalence of low back complaints, that is, considerably more than in our sample. Of particular note, we found that 6%–9% of the participants had daily musculoskeletal complaints and that 3%–7% had been forced to change work tasks or jobs due to their complaints. A history of pain is strongly associated with current pain,[38][39] and studies—however, mainly among blue-collar workers—have suggested uninterrupted, severe neck and shoulder pain over a long period of time to be associated with decreased workability and increased sick leave.[40]

We did not find any general association between lean maturity and musculoskeletal complaints. However, cross-sectional analyses of each of the 4P showed that philosophy was positively associated with neck complaints, while people and partners had a negative association. In the longitudinal analysis, none of the 4P were associated with complaints in any of the four investigated body regions, except for people and partners, which was positively associated with neck complaints when the other 4P were controlled for and complaints at T1 were included as a covariate. These results are to some extent difficult to interpret, but they can be explained by earlier findings suggesting that both physical and psychosocial working conditions can be influenced by lean maturity and are risk factors for musculoskeletal complaints.[17][31] Since very few studies have been conducted in primary care, we cannot compare our results to previous reports from that sector. A comparison of our results with studies conducted in a hospital setting is less relevant, due to major differences in physical and psychosocial working conditions between working in a hospital[20][23–29] and in primary care.[31]

Also, our results differ from studies of other occupational sectors than healthcare, where increased lean has been identified as a risk factor for musculoskeletal complaints.[17–19] Likely, the effect of lean depends on the occupational sector, inasmuch as both working conditions and implementations of lean will differ according to the occupational context. Thus, our results are difficult to compare with previous studies because they may have used other descriptions of lean and have investigated other ways of implementing lean than those pertinent to primary care.[1][4] We measured lean maturity using the LiHcQ, which addresses all of Liker's 14 lean principles,[5] while most of the studies included in Mazzocato et al's[10] review of lean in healthcare assessed lean to a less comprehensive level. Another important aspect in the context of previous papers is the risk of publication bias, that is, that published studies mainly report significant results, while leaving out non-significant results.[41] Our results indicate that musculoskeletal complaints among primary care staff are a concern, especially as some staff members have daily complaints and some have to change jobs or work tasks due to these complaints.

We did not find any clear associations between lean maturity and musculoskeletal complaints, but we

**Table 5** Cross-sectional associations between lean maturity and musculoskeletal complaints in the shoulders, neck, low back and upper back

| | Model 1a | | | | Model 1b | | | |
|---|---|---|---|---|---|---|---|---|
| | β | 95% CI | P value | OR range; MI data set 1–5 | β | 95% CI | P value | OR range; MI data set 1–5 |
| **Shoulder complaints** | Intercept –0.61 | | | | Intercept –0.80 | | | |
| Lean maturity in total | –0.02 | –0.04; 0.01 | 0.17 | 0.98; 0.99 | | | | |
| Philosophy | | | | | 0.12 | –0.02; 0.26 | 0.09 | 1.09; 1.17 |
| Processes | | | | | 0.01 | –0.13; 0.15 | 0.89 | 0.97; 1.09 |
| People and partners | | | | | –0.16 | –0.34; 0.03 | 0.10 | 0.83; 0.89 |
| Problem solving | | | | | –0.08 | –0.33; 0.17 | 0.49 | 0.81; 0.98 |
| **Control variables** | | | | | | | | |
| Female gender | 0.91 | 0.13; 1.69 | 0.02 | 2.32; 2.62 | 0.88 | 0.07; 1.69 | 0.03 | 2.20; 2.59 |
| BMI | 0.03 | –0.05; 0.10 | 0.51 | 1.02; 1.04 | 0.03 | –0.06; 0.11 | 0.56 | 1.02; 1.03 |
| Age 43–51 years | 0.20 | –0.56; 0.97 | 0.61 | 1.15; 1.37 | 0.23 | –0.58; 1.04 | 0.58 | 1.13; 1.43 |
| 52–57 years | 0.74 | –0.20; 1.67 | 0.12 | 2.02; 2.14 | 0.85 | –0.16; 1.86 | 0.10 | 2.13; 2.80 |
| 58 years and older | –0.22 | –0.10; 0.55 | 0.58 | 0.77; 0.86 | –0.22 | –1.04; 0.59 | 0.59 | 0.76; 0.86 |
| **Neck complaints** Intercept –0.34 | | | | | Intercept –0.60 | | | |
| Lean maturity in total | –0.01 | –0.03; 0.01 | 0.50 | 0.99; 1.00 | | | | |
| Philosophy | | | | | 0.19 | 0.07; 0.31 | <0.01 | 1.19; 1.23 |
| Processes | | | | | –0.02 | –0.13; 0.08 | 0.66 | 0.95; 1.00 |
| People and partners | | | | | –0.19 | –0.36; –0.02 | 0.03 | 0.78; 0.85 |
| Problem solving | | | | | –0.00 | –0.22; 0.21 | 0.97 | 0.93; 1.06 |
| **Control variables** | | | | | | | | |

Continued

**Table 5** Continued

| | Model 1a | | | | Model 1b | | | |
|---|---|---|---|---|---|---|---|---|
| | β | 95% CI | P value | OR range; MI data set 1–5 | β | 95% CI | P value | OR range; MI data set 1–5 |
| Female gender | 1.02 | −0.05; 2.09 | 0.06 | 2.60; 3.05 | 1.06 | −0.07; 2.18 | 0.07 | 2.51; 3.36 |
| BMI | 0.02 | −0.04; 0.08 | 0.55 | 1.01; 1.03 | 0.03 | −0.04; 0.09 | 0.41 | 1.02; 1.04 |
| Age 43–51 years | −0.07 | −0.81; 0.67 | 0.85 | 0.92; 0.95 | −0.12 | −0.88; 0.64 | 0.76 | 0.84; 0.92 |
| 52–57 years | −0.90 | −1.65; −0.15 | 0.02 | 0.39; 0.43 | −0.94 | −1.74; −0.13 | 0.02 | 0.36; 0.46 |
| 58 years and older | −0.97 | −1.77; −0.17 | 0.02 | 0.36; 0.40 | −1.05 | −1.93; −0.18 | 0.02 | 0.32; 0.37 |
| **Low back complaints** Intercept −1.44 | | | | | Intercept −1.59 | | | |
| Lean maturity in total | −0.02 | −0.04; 0.00 | 0.09 | 0.98; 0.98 | | | | |
| Philosophy | | | | | 0.08 | −0.06; 0.21 | 0.29 | 1.06; 1.12 |
| Processes | | | | | 0.00 | −0.11; 0.12 | 0.97 | 0.95; 1.04 |
| People and partners | | | | | −0.14 | −0.33; 0.05 | 0.14 | 0.83; 0.92 |
| Problem solving | | | | | −0.05 | −0.29; 0.19 | 0.66 | 0.89; 1.07 |
| **Control variables** | | | | | | | | |
| Female gender | 0.28 | −0.56; 1.12 | 0.51 | 1.18; 1.66 | 0.26 | −0.65; 1.17 | 0.57 | 1.08 1.66 |
| BMI | 0.08 | 0.01; 0.15 | 0.02 | 1.08; 1.09 | 0.08 | 0.01; 0.16 | 0.03 | 1.07; 1.10 |
| Age 43–51 years | 0.39 | −0.43; 1.20 | 0.35 | 1.34; 1.59 | 0.40 | −0.40; 1.20 | 0.33 | 1.38; 1.55 |
| 52–57 years | −0.28 | −1.20; 0.65 | 0.56 | 0.58; 0.88 | −0.21 | −1.24; 0.83 | 0.69 | 0.56; 0.98 |
| 58 years and older | −0.22 | −0.96; 0.52 | 0.56 | 0.68; 0.91 | −0.22 | −0.99; | 0.58 | 0.67; 0.92 |
| **Upper back complaints** Intercept −1.96 | | | | | Intercept −2.01 | | | |

Continued

**Table 5** Continued

| | Model 1a | | | | Model 1b | | | |
|---|---|---|---|---|---|---|---|---|
| | β | 95% CI | P value | OR range; MI data set 1–5 | β | 95% CI | P value | OR range; MI data set 1–5 |
| **Lean maturity in total** | 0.00 | −0.02; 0.02 | 0.96 | 1.00; 1.00 | | | | |
| **Philosophy** | | | | | 0.01 | −0.11; 0.14 | 0.82 | 0.96; 1.05 |
| **Processes** | | | | | 0.02 | −0.09; 0.12 | 0.77 | 0.98; 1.03 |
| **People and partners** | | | | | 0.04 | −0.14; 0.23 | 0.65 | 1.01; 1.10 |
| **Control variables** | | | | | | | | |
| **Female gender** | 0.75 | −0.21; 1.70 | 0.13 | 1.88; 2.34 | 0.69 | −0.29; 1.67 | 0.17 | 1.78; 2.27 |
| **BMI** | 0.03 | −0.04; 0.10 | 0.40 | 1.02; 1.03 | 0.03 | −0.04; 0.10 | 0.44 | 1.02; 1.04 |
| **Age 43–51 years** | −0.25 | −0.89; 0.40 | 0.46 | 0.74; 0.85 | −0.22 | −0.89; 0.45 | 0.52 | 0.76; 0.86 |
| **52–57 years** | −0.10 | −0.99; 0.79 | 0.83 | 0.83; 1.01 | −0.05 | −0.98; 0.88 | 0.92 | 0.87; 1.06 |
| **58 years and older** | −0.43 | −1.21; 0.36 | 0.28 | 0.60; 0.71 | −0.40 | −1.21; 0.41 | 0.33 | 0.60; 0.74 |

Reference in the control variables were male and age ≤42 years.
BMI, body mass index; MI, multiple imputation; 4P, philosophy, processes, people and partners, and problem solving; β, unstandardised beta coefficient.

**Table 6** Association between lean maturity and musculoskeletal complaints in the shoulders over time

| Shoulder complaints at T2 | Model 2a, only lean maturity in total and covariates | | | | Model 2b, complaints T1 added | | | | Model 2c, each of the 4P | | | | Model 2d, complaints T1 added | | | |
|---|---|---|---|---|---|---|---|---|---|---|---|---|---|---|---|---|
| | β | 95%CI | P value | OR range; MI data set 1–5 | β | 95%CI | P value | OR range; MI data set 1–5 | β | 95%CI | P value | OR range; MI data set 1–5 | β | 95%CI | P value | OR range; MI data set 1–5 |
| Intercept | −0.29 | | | | −1.02 | | | | −0.43 | | | | −1.12 | | | |
| Lean maturity in total at T1 | −0.002 | −0.03; 0.02 | 0.85 | 0.996; 1.00 | 0.006 | −0.02; 0.03 | 0.61 | 1.00; 1.10 | | | | | | | | |
| Shoulder complaints at T1 | | | | | 2.00 | 1.36; 2.66 | <0.01 | 6.83; 8.07 | | | | | | | | |
| Philosophy | | | | | | | | | 0.07 | −0.05; 0.19 | 0.23 | 1.06; 1.09 | 0.03 | −0.11; 0.16 | 0.70 | 1.05; 1.01 |
| Processes | | | | | | | | | −0.03 | −0.13; 0.08 | 0.64 | 0.95; 1.01 | −0.04 | −0.14; 0.07 | 0.51 | 0.95; 1.00 |
| People and partners | | | | | | | | | −0.01 | −0.18; 0.15 | 0.88 | 0.96; 1.01 | 0.07 | −0.11; 0.25 | 0.46 | 1.04; 1.10 |
| Problem- solving | | | | | | | | | −0.02 | −0.19; 0.16 | 0.85 | 0.92; 1.03 | 0.02 | −0.13; 0.17 | 0.80 | 0.98; 1.06 |
| Complaints at T1 | | | | | | | | | | | | | 2.02 | 1.36; 2.68 | <0.01 | 6.72; 8.22 |
| Female gender | 0.29 | −0.26; 0.83 | 0.30 | 1.31; 1.36 | −0.14 | −0.69; 0.41 | 0.61 | 0.83; 0.93 | 0.27 | −0.28; 0.82 | 0.34 | 1.22; 1.36 | −0.14 | −0.72; 0.43 | 0.62 | 0.81; 0.96 |
| BMI | 0.02 | −0.05; 0.08 | 0.63 | 1.00; 1.03 | 0.01 | −0.06; 0.07 | 0.85 | 0.99; 1.02 | 0.02 | −0.05; 0.09 | 0.55 | 1.00; 1.04 | 0.01 | −0.06; 0.08 | 0.69 | 0.99; 1.03 |
| Age 43–51 years | 0.03 | −0.74; 0.80 | 0.94 | 0.96; 1.11 | −0.08 | −0.98; 0.82 | 0.87 | 0.96; 1.01 | 0.03 | −0.76; 0.82 | 0.942 | 0.95; 1.09 | −0.08 | −1.00; 0.83 | 0.86 | 0.87; 1.00 |
| 52–57 years | 0.34 | −0.52; 1.20 | 0.44 | 1.29; 1.51 | 0.02 | −0.92; 0.96 | 0.96 | 0.92; 1.12 | 0.34 | −0.56; 1.24 | 0.456 | 1.24; 1.48 | −0.03 | −0.99; 0.94 | 0.96 | 0.86; 1.05 |
| 58 years and older | −0.26 | −0.94; 0.43 | 0.47 | 0.73; 0.84 | −0.20 | −0.86; 0.46 | 0.56 | 0.76; 0.92 | −0.27 | −0.98; 0.44 | 0.454 | 0.71; 0.86 | −0.21 | −0.87; 0.45 | 0.52 | 0.76; 0.90 |

Predictors were lean maturity in total at T1 (models 2a, 2b), or each of the 4Ps (models 2c, 2d); complaints at T2 were the outcome in all models. Gender, BMI and age were included as covariates in all models.

Reference in the control variables were male and age ≤42 year.

BMI, body mass index; MI, multiple imputation; 4P, philosophy, processes, people and partners, and problem solving; T1, time 1; T2, time 2; β, unstandardised beta coefficient.

**Table 7** Association between lean maturity and musculoskeletal complaints in the neck over time

| Neck complaints at T2 | Model 2a, only lean maturity in total and covariates | | | | Model 2b, complaints T1 added | | | | Model 2c, 4P (instead of lean maturity in total) | | | | Model 2d, complaints T1 added | | | |
|---|---|---|---|---|---|---|---|---|---|---|---|---|---|---|---|---|
| | β | 95%CI | P value | OR range; MI data set 1–5 | β | 95%CI | P value | OR range; MI data set 1–5 | β | 95%CI | P value | OR range; MI data set 1–5 | β | 95%CI | P value | OR range; MI data set 1–5 |
| Intercept | −0.36 | | | | −1.53 | | | | −0.60 | | | | −1.74 | | | |
| Lean maturity in total at T1 | 0.006 | −0.01; 0.03 | 0.57 | 1.00; 1.01 | 0.01 | −0.01; 0.04 | 0.28 | 1.01; 1.02 | | | | | | | | |
| Neck complaints at T1 | | | | | 2.31 | 1.66; 2.96 | <0.01 | 9.54; 10.71 | | | | | | | | |
| Philosophy | | | | | | | | | 0.12 | −0.02; 0.25 | 0.08 | 1.09; 1.15 | 0.03 | −0.10; 0.17 | 0.63 | 1.00; 1.06 |
| Processes | | | | | | | | | −0.07 | −0.16; 0.03 | 0.15 | 0.92; 0.96 | −0.07 | −0.20; 0.06 | 0.25 | 0.90; 0.97 |
| People and partners | | | | | | | | | 0.04 | −0.12; 0.20 | 0.64 | 1.00; 1.07 | 0.17 | 0.02; 0.33 | 0.03 | 1.16; 1.22 |
| Problem solving | | | | | | | | | 0.03 | −0.12; 0.18 | 0.69 | 1.00; 1.06 | 0.04 | −0.16; 0.24 | 0.68 | 0.98; 1.10 |
| Neck complaints at T1 | | | | | | | | | | | | | 2.40 | 1.74; 3.03 | <0.01 | 10.40; 11.40 |
| Female gender | 0.84 | 0.02; 1.66 | 0.04 | 2.28; 2.35 | 0.52 | −0.12; 1.15 | 0.11 | 1.58; 1.81 | 0.84 | 0.03; 1.66 | 0.04 | 2.30; 2.39 | 0.49 | −0.20; 1.18 | 0.17 | 1.48; 1.76 |
| BMI | −0.01 | −0.07; 0.04 | 0.67 | 0.98; 1.00 | −0.03 | −0.09; 0.04 | 0.39 | 0.96; 1.00 | 0.003 | −0.05; 0.06 | 0.90 | 0.99; 1.02 | −0.01 | −0.08; 0.06 | 0.76 | 0.97; 1.01 |
| Age 43–51 years | 0.16 | −0.58; 0.89 | 0.674 | 1.08; 1.23 | 0.29 | −0.63; 1.21 | 0.54 | 1.19; 1.43 | 0.13 | −0.61; 0.86 | 0.74 | 1.08; 1.18 | 0.26 | −0.68; 1.19 | 0.59 | 1.17; 1.39 |
| 52–57 years | −0.03 | −0.80; 0.75 | 0.95 | 0.90; 1.07 | 0.57 | −0.39; 1.53 | 0.25 | 1.60; 1.86 | −0.12 | −0.92; 0.69 | 0.78 | 0.79; 0.99 | 0.44 | −0.58; 1.46 | 0.39 | 1.34; 1.71 |
| 58 years and older | −0.54 | −1.22; 0.14 | 0.12 | 0.56; 0.63 | −0.04 | −0.82; 0.74 | 0.92 | 0.87; 1.12 | −0.61 | −1.32; 0.10 | 0.09 | 0.51; 0.59 | −0.10 | −0.91; 0.71 | 0.80 | 0.79; 1.06 |

Predictors were lean maturity in total at T1 (models 2a and 2b) or each of 4Ps (model 2c and 2d); complaints at T2 were the outcome in all models. Gender, BMI and age were included as covariates in all models.
Reference groups in the control variables were male and age ≤42 years.
BMI, body mass index; MI, multiple imputation; 4P, philosophy, processes, people and partners, and problem solving; T1, time 1; T2, time 2; β, unstandardised beta coefficient.

**Table 8** Association between lean maturity and musculoskeletal complaints in the low back over time

| Low back complaints at T2 | Model 2a, only lean maturity in total and covariates | | | | Model 2b, complaints T1 added | | | | Model 2c, 4P (instead of lean maturity in total) | | | | Model 2d, complaints T1 added | | | |
|---|---|---|---|---|---|---|---|---|---|---|---|---|---|---|---|---|
| | β | 95% CI | P value | OR range; MI data set 1–5 | β | 95% CI | P value | OR range; MI data set 1–5 | β | 95% CI | P value | OR range; MI data set 1–5 | β | 95% CI | P value | OR range; MI data set 1–5 |
| Intercept | −1.34 | | | | −1.89 | | | | −1.60 | | | | −2.20 | | | |
| Lean maturity in total at T1 | 0.004 | −0.02; 0.03 | 0.74 | 1.00; 1.01 | 0.02 | −0.01; 0.04 | 0.26 | 1.01; 1.02 | | | | | | | | |
| Low back complaints at T1 | | | | | 2.00 | 1.44; 2.55 | <0.01 | 6.46; 8.64 | | | | | | | | |
| Philosophy | | | | | | | | | 0.11 | −0.03; 0.25 | 0.13 | 1.09; 1.17 | 0.10 | −0.07; 0.27 | 0.25 | 1.07; 1.18 |
| Processes | | | | | | | | | −0.05 | −0.14; 0.04 | 0.25 | 0.93; 0.97 | −0.07 | −0.19; 0.05 | 0.24 | 0.90; 0.97 |
| People and partners | | | | | | | | | 0.06 | −0.15; 0.26 | 0.59 | 1.02; 1.16 | 0.15 | −0.09; 0.39 | 0.21 | 1.11; 1.28 |
| Problem solving | | | | | | | | | −0.02 | −0.17; 0.13 | 0.77 | 0.95; 1.00 | 0.001 | −0.17; 0.17 | 0.99 | 0.97; 1.03 |
| Complaints at T1 | | | | | | | | | | | | | 2.07 | 1.50; 2.64 | <0.01 | 6.77; 9.43 |
| Female gender | 0.53 | −0.32; 1.37 | 0.22 | 1.51; 1.92 | 0.50 | −0.51; 1.50 | 0.33 | 1.37; 2.07 | 0.48 | −0.37; 1.33 | 0.27 | 1.46; 1.81 | 0.46 | −0.57; 1.48 | 0.38 | 1.28; 2.00 |
| BMI | 0.02 | −0.05; 0.10 | 0.53 | 1.01; 1.04 | −0.01 | −0.09; 0.07 | 0.74 | 0.97; 1.00 | 0.04 | −0.04; 0.11 | 0.31 | 1.03; 1.05 | 0.004 | −0.07; 0.08 | 0.91 | 0.99; 1.02 |
| Age 43–51 years | 0.38 | −0.28; 1.03 | 0.26 | 1.36; 1.54 | 0.26 | −0.40; 0.92 | 0.44 | 1.25; 1.36 | 0.37 | −0.32; 1.05 | 0.29 | 1.36; 1.52 | 0.24 | −0.47; 0.94 | 0.51 | 1.23; 1.32 |
| 52–57 years | 0.18 | −0.61; 0.96 | 0.66 | 1.15; 1.25 | 0.38 | −0.41; 1.17 | 0.35 | 1.35; 1.76 | 0.13 | −0.71; 0.97 | 0.76 | 1.08; 1.25 | 0.28 | −0.61; 1.18 | 0.53 | 1.16; 1.67 |
| 58 years and older | −0.13 | −0.72; 0.46 | 0.66 | 0.81; 0.96 | −0.04 | −0.68; 0.60 | 0.91 | 0.88; 1.09 | −0.17 | −0.79; 0.44 | 0.58 | 0.78; 0.94 | −0.08 | −0.79; 0.63 | 0.82 | 0.80; 1.04 |

Predictors were lean maturity in total at T1 (models 2a and 2b) or each of 4Ps (model 2c and 2d); complaints at T2 were the outcome in all models. Gender, BMI and age were included as covariates in all models.
Reference groups in the control variables were male and age ≤42 years.
BMI, body mass index; MI, multiple imputation; 4P, philosophy, processes, people and partners, and problem solving; T1, time 1; T2, time 2; β, unstandardised beta coefficient.

**Table 9** Association between lean maturity and musculoskeletal complaints in the upper back over time

| Upper back complaints at T2 | Model 2a, only lean maturity in total and covariates | | | | Model 2b, complaints T1 added | | | | Model 2c, 4P (instead of lean maturity in total) | | | | Model 2d, complaints T1 added | | | |
|---|---|---|---|---|---|---|---|---|---|---|---|---|---|---|---|---|
| | β | 95% CI | P value | OR range; MI data set 1–5 | β | 95% CI | P value | OR range; MI data set 1–5 | β | 95% CI | P value | OR range; MI data set 1–5 | β | 95% CI | P value | OR range; MI data set 1–5 |
| Intercept | −0.74 | | | | −0.91 | | | | −0.82 | | | | −0.94 | | | |
| Lean maturity in total at T1 | 0.002 | −0.02; 0.02 | .88 | 1.00; 1.01 | 0.003 | −0.02; 0.03 | .84 | 1.00; 1.01 | | | | | | | | |
| Upper back complaints at T1 | | | | | 2.17 | 1.43; 2.91 | <0.01 | 7.44; 9.50 | | | | | | | | |
| Philosophy | | | | | | | | | 0.01 | −0.14; 0.16 | .86 | 0.95; 1.08 | 0.008 | −0.16; 0.18 | .93 | 0.96; 1.08 |
| Processes | | | | | | | | | −0.04 | −0.14; 0.06 | .40 | 0.94; 0.98 | −0.07 | −0.20; 0.07 | .32 | 0.90; 0.98 |
| People and partners | | | | | | | | | 0.13 | −0.10; 0.36 | .25 | 1.09; 1.24 | 0.15 | −0.12; 0.42 | .26 | 1.07; 1.32 |
| Problem solving | | | | | | | | | −0.03 | −0.21; 0.15 | .78 | 0.95; 0.99 | 0.01 | −0.20; 0.23 | .89 | 0.98; 1.06 |
| Complaints at T1 | | | | | | | | | | | | | 2.21 | 1.42; 3.01 | <0.01 | 7.57; 10.56 |
| Female gender | −0.004 | −0.75; 0.74 | .99 | 0.82; 1.15 | −0.41 | −1.39; 0.57 | 0.41 | 0.53; 0.78 | −0.05 | −0.79; 0.70 | 0.90 | 0.78; 1.11 | −0.44 | −1.42; 0.55 | 0.38 | 0.52; 0.77 |
| BMI | 0.002 | −0.07; 0.07 | .96 | 0.99; 1.03 | −0.02 | −0.10; 0.066 | .71 | 0.96; 1.01 | 0.01 | −0.07; 0.09 | .79 | 0.99; 1.04 | −0.01 | −0.10; 0.09 | .91 | 0.97; 1.03 |
| Age 43–51 years | 0.14 | −0.59; 0.87 | .70 | 0.96; 1.39 | 0.32 | −0.55; 1.19 | .46 | 1.04; 1.84 | 0.15 | −0.62; 0.92 | .69 | 0.97; 1.42 | 0.34 | −0.58; 1.26 | .46 | 1.05; 1.85 |
| 52–57 years | −0.27 | −1.13; 0.59 | .54 | 0.72; 0.86 | −0.29 | −1.38; 0.81 | .61 | 0.65; 0.85 | −0.31 | −1.19; 0.57 | .49 | 0.67; 0.84 | −0.36 | −1.53; 0.81 | .55 | 0.59; 0.85 |
| 58 years and older | −0.65 | −1.35; 0.05 | .07 | 0.47; 0.59 | −0.57 | −1.38; 0.24 | .17 | 0.49; 0.68 | −0.67 | −1.39; 0.05 | .07 | 0.46; 0.58 | −0.60 | −1.47; 0.27 | .18 | 0.46; 0.68 |

Predictors were lean maturity in total at T1 (models 2a and 2b) or each of 4Ps (model 2c and 2d); complaints at T2 were the outcome in all models. Gender, BMI and age were included as covariates in all models.
Reference groups in the control variables were male and age ≤42 years.
BMI, body mass index; MI, multiple imputation; 4P, philosophy, processes, people and partners, and problem solving; T1, time 1; T2, time 2; β, unstandardised beta coefficient.

encourage future research to assess both lean maturity and musculoskeletal complaints over a more extended period of time than 1 year. Further research will allow for a deeper understanding of whether changes in lean maturity and musculoskeletal complaints are associated, and if so, what elements of lean that are particularly important risk factors.

## Strengths and limitations

One limitation of the study is the use of a convenience sample and the relatively low response rate[39]; this limits the generalisability of our results. Using staff reports of Lean maturity and musculoskeletal complaints can also be viewed as a limitation, as responses may suffer from common-method bias related to the participants' attitudes towards their workplace, their general health, their tendency to give socially desirable answers and inaccurate recall. However, staff members can be considered to have the best insights into their own health and working conditions. Our sample comprised mainly female registered nurses. However, this reflected well the composition of primary care staff in Sweden, where the most common profession is registered nurses,[42] and even healthcare in general, where 88% of the registered nurses are women.[43] Data concerning activities conducted by the primary care units to adopt lean between T1 and T2 were not collected, which limits the interpretation of our results. A strength of the study was its longitudinal design and the use of valid and, in general, consistent instruments for measuring lean maturity[33] as well as complaints. An additional strength was the MI approach for handling missing data, since this procedure does not lead to loss of data, such as several other alternatives. However, a major limitation of our study is the follow-up time frame of only 1 year. It may be too short to identify changes in both musculoskeletal health and lean maturity (figure 2).

## Conclusions

The prevalence of musculoskeletal complaints among primary care staff was high and did not change during 1 year. The extent of lean maturity was not associated with complaints, neither in cross-sectional analyses nor in a 1-year predictive analysis.

**Acknowledgements** We would like to thank all participants in the study and their primary healthcare institutions.

**Contributors** The study was designed by all authors. Data were collected by MK and analysed by MK, HH and ME. MK was primarily responsible for drafting the manuscript, which was critically revised by SEM, LB, HH and ME. All authors read and approved the final manuscript. MK acting as a guarantor.

**Funding** This research was financially supported by the University of Gävle (no grant number).

**Disclaimer** The funder had no role in the study design, data collection, analysis, data interpretation or the decision to submit the paper for publication.

**Competing interests** None declared.

**Patient and public involvement** Patients and/or the public were involved in the design, or conduct, or reporting, or dissemination plans of this research. Refer to the Methods section for further details.

**Patient consent for publication** Not applicable.

**Ethics approval** The study was approved by the Regional Ethical Review Board in Uppsala (Reg. no. 2014/525 and 2014/525/1). Participants gave informed consent to participate in the study before taking part.

**Provenance and peer review** Not commissioned; externally peer reviewed.

**Data availability statement** The data presented in this study are available from the corresponding author on reasonable request. Individual data are not publicly available due to general data protection regulations and personal data ordinance.

**ORCID iD**
Monica Kaltenbrunner http://orcid.org/0000-0003-2211-620X

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
