## [Reviewer comments · BMJ Open]

ARTICLE DETAILS

TITLE (PROVISIONAL)	Associations between Lean maturity in primary care and musculoskeletal complaints among staff: a longitudinal study
AUTHORS	Kaltenbrunner, Monica; Kaltenbrunner, Monica; Mathiassen, Svend; Bengtsson, Lars; Högberg, Hans; Engström, Maria

VERSION 1 – REVIEW

REVIEWER	Oakman, Jodi La Trobe University, Public Health
REVIEW RETURNED	31-Mar-2022

GENERAL COMMENTS	Thank you for the opportunity to review this paper. Overall, it is a well written paper and investigates an interesting and worthwhile topic. Some comments though to improve the paper prior to it being ready for publication. Abstract Keywords: why is Liker included here, this is an author's name, not usual practice for key words Introduction The introduction needs to have some paragraphs inserted; the first page is one entire paragraph which makes it somewhat challenging for the readers. There are issues that are not as well considered as they could be in relation to the huge body of literature that relates to the multifactorial aetiology of MSP. As a result, there is a disconnect when the authors talk about the risk factors for MSP and the causes, which are all listed as physical but the components of lean that have been introduced relate to the organisation of work, which are also significant risk factors for MSP but these are not mentioned here. This somewhat undermines the rationale for the study raising questions about why would lean be relevant. I understand the logic, but this is not clear on paper.
---

	I am not sure the key issue of relevance to underpin the study need is that Swedish primary care has not been done, that is not a gap that makes the study worthwhile; it is the linking of the relevance of lean and the subsequent impact. On page 8 the authors then state again that lean sets the scene for working conditions, but again this is not reflected in the references or the explanation about what are the causal factors for MSP that relates to the important psychosocial factors, this is a key issue that needs to be addressed to ensure that the logic for undertaking the study is clear. Method, I find the longitudinal, descriptive, and correlational design statement somewhat challenging and particularly in that order, because it does not follow the logical order in which the analysis was undertaken. It is not clear whether Lean was measured at T2, I understand why one might consider that the organisational maturity at a point in time influences pain over time. But, a key issue here that is not well discussed is the issue of organisational stability and the maintenance of Lean, which is potentially and impact here, preferable to have it measured at two time points, but it is a limitation that needs to be discussed but links with the limited discussion about organisational issue which are the centre of Lean and so need to be brought into the paper to add clarity about the impact of Lean on individuals. Line 37: minor wording issue, Should be: Units were required to have adopted Lean to some extent to be eligible for participation in the study. Participants 46: This issue is not a technical issue but just something that is annoying to those of us who reside in the southern hemisphere
--	--

where we use months of the year rather than seasons, which is more accurate. In current form this could be any of six months of the year (spring in the south and the north), it is more accurate to use dates than seasons. I am aware that people commonly do this, but it is a little insensitive. As I said not a scientific issue but rather one of pragmatism and adopting a more inclusive approach.

Page 9 How many paper versions were sent and returned?

Discussion

The relationship between people and neck pain, is stated to be difficult to interpret. I am not sure why. And without seeing the items somewhat challenging to understand but this relates to the comments on the introduction and the limited focus on other causal factors in the development of MSP. This relationship is widely reported and so would be expected, that it is within the context of LEAN is not the issue here but that it supports previous work that has identified a full range of psychosocial factors that are related to neck pain and in particular females.

So this is an interesting finding but I don't accept that it is actually difficult to explain. I say that in the context I do not have the questions but guessing from my understanding of lean that this is the sort of thing that is covered. So I do not agree with line 36 that this requires further explanation, what is required is better understanding of the interventions that address these issues.

And that you are seeking a population of primary care is somewhat narrow, what about health care or other populations

	where these relationships are well established. And you population was mainly nurses so this can be compared but some caveats of course. Line 40, it is not clear what this means, our results differ..... If it is only suggested in other sectors, maybe this needs rewording so that it is clearer that other studies have identified lean as associated with increased MSP risk. Line 48 this is very true, in relation to occupational context but this is why it is not clear why the results are surprising that the "people" variable was significant. Page 19 Line 22 is should be are. Line 35-37 needs rewriting for clarity, too many commas, and unclear Line 44 remove which even, and just say had a relatively low response rate. Some mention of why you would not expect a change in MSP over one year would be useful, but also why you might expect a change in Lean measures, even if this was not done but it is a clear limitation, that is how sustainable were the Lean initiatives. Some further explanation of what was done might actually help with interpretation of the results because if Lean is considered the intervention, then to what extent was the implementation successful. I realise this is not the intent of the article but nonetheless it does then provide some context as to what happened or did not happen and therefore the potential influence on the results. Minor comment for the start of the discussion : the use of the term prevalence's needs reviewing, used twice in the first three lines. Could be The musculoskeletal prevalence reporting the current study is higher than in the general or similar.
--	---

REVIEWER	Hoe, Victor
----------	-------------

	University of Malaya, Kuala Lumpur
REVIEW RETURNED	02-Apr-2022

GENERAL COMMENTS	Thank you for the opportunity to review your paper titled “Associations between Lean Maturity in primary care and musculoskeletal complaints among staff. A longitudinal study”. The study reported findings from a longitudinal study among healthcare workers from Sweden. The study assesses the implementation of Lean management and its association with musculoskeletal complaints. The authors focus on four sites with the highest MSD complaints. The study found that there was no significant association between Lean maturity and MSD at the various sites in the cross-sectional and longitudinal analysis. Overall, the manuscript has been well written, and it is easy to read and understand. The clarity could be improved further by explaining in more detail the concept of Lean in Health care. It would be good also to provide a background on the history of Lean implementation in Sweden. There is one major issue that I am not clear about. What is the association between Lean and MSD? There is no clear pathophysiological linkage between Lean and MSD, especially when we are focusing on localized MSD or MSD at a single site. The risk factors for localized MSDs are different from generalized MSD, i.e., multisite MSDs. Most of the time localized MSDs is the result of issues related to physical ergonomics issue, e.g., bad posture, repetitive movement, etc. If the authors are focusing on multisite MSD then the linkage between Lean and MSDs may be more apparent. Multisite MSDs are found to be associated by psychological, organizational and management styles. I would suggest if possible for the author to conduct an analysis looking at multisite pain to see if there are any associations.
---

VERSION 1 – AUTHOR RESPONSE

Reviewer: 1

Dr. Jodi Oakman, La Trobe University

Keywords: why is Liker included here, this is an author’s name, not usual practice for key words

Response: You are right, it is not usual practice. There are many different descriptions of Lean, and in our case, we think it is important to include Liker as a Keyword as it makes it easier to find our article and also clarifies that our study is based on Liker’s description of Lean.

The introduction needs to have some paragraphs inserted; the first page is one entire paragraph which makes it somewhat challenging for the readers.

Response: Yes, you are correct, thank you for highlighting this. This is now adjusted.

There are issues that are not as well considered as they could be in relation to the huge body of literature that relates to the multifactorial aetiology of MSP. As a result, there is a disconnect when the authors talk about the risk factors for MSP and the causes, which are all listed as physical but the components of lean that have been introduced relate to the organisation of work, which are also significant risk factors for MSP but these are not mentioned here. This somewhat undermines the

rationale for the study raising questions about why would lean be relevant. I understand the logic, but this is not clear on paper.

Response: Thank you for underlining the lack of clarity. We have rearranged and added some text at page 5, line 99-102, page 5, line 115-117, page 5-6, line 120-127, page 15-16, line 335-338, aiming to make the link between Lean, working conditions and MSD clearer for the reader; including both physical and psychosocial factors of relevance in a Lean context.

I am not sure the key issue of relevance to underpin the study need is that Swedish primary care has not been

done, that is not a gap that makes the study worthwhile; it is the linking of the relevance of lean and the subsequent impact.

On page 8 the authors then state again that lean sets the scene for working conditions, but again this is not reflected in the references or the explanation about what are the causal factors for MSP that relates to the important psychosocial factors, this is a key issue that needs to be addressed to ensure that the logic for undertaking the study is clear.

Response: The description of a general link is now included at page 5, line 99-102 and page 5-6 line 115- 127 where we tried to make it clear that Lean influence both physical and psychosocial working conditions and that working conditions are an issue in primary care.

I find the longitudinal, descriptive, and correlational design statement somewhat challenging and particularly in that order, because it does not follow the logical order in which the analysis was undertaken.

Response: We have changed it to a 'descriptive, correlational, and longitudinal design'. Page 2, line 27. Page 6, line 30.

It is not clear whether Lean was measured at T2, I understand why one might consider that the organizational maturity at a point in time influences pain over time. But, a key issue here that is not well discussed is the issue of organisational stability and the maintenance of Lean, which is potentially and impact here, preferable to have it measured at two time points, but it is a limitation that needs to be discussed but links with the limited discussion about organisational issue which are the centre of Lean and so need to brought into the paper to add clarity about the impact of Lean on individuals.

Response: We have, indeed, measured Lean maturity at both T1 and T2. At page 2, line 29-32, we mention this the first time, thereafter we mention it again, for instance at page 2, line 40-41, page 7, line 154 and line 155-158.

We have also included a figure (figure 2) which reports the change in Lean between T1 and T2.

In addition, we have now added a sentence at page 8, line 193-194 to make it clearer that Lean is measured twice.

Line 37: minor wording issue,

Should be: Units were required to have adopted Lean to some extent to be eligible for participation in the study.

Response: This is now adjusted at page 7, line 148-150.

Participants

46: In current form this could be any of six months of the year (spring in the south and the north), it is more accurate to use dates than seasons.

Response: The recruitment month is now added instead of the term 'spring' on Page 7, line 153-154.

Page 9

How many paper versions were sent and returned?

Response: Unfortunately, this information was not available to us and therefore cannot be reported. The primary care units had the answer for distributing the paper versions of the questionnaire and we did not receive any information about the success rate in doing that.

Discussion

The relationship between people and neck pain, is stated to be difficult to interpret. I am not sure why. And without seeing the items somewhat challenging to understand but this relates to the comments on the introduction and the limited focus on other causal factors in the development of MSP. This relationship is widely reported and so would be expected, that it is within the context of LEAN is not the issue here but that it supports previous work that has identified a full range of psychosocial factors that are related to neck pain and in particular females. So this is an interesting finding but I don't accept that it is actually difficult to explain. I say that in the context I do not have the questions but guessing from my understanding of lean that this is the sort of thing that is covered. So I do not agree with line 36 that this requires further explanation, what is required is better understanding of the interventions that address these issues.

Response: We have made the interpretation of our results in the context of relationships between working conditions, Lean and musculoskeletal complaints clearer at page 15, line 335-338.

And that you are seeking a population of primary care is somewhat narrow, what about health care or other populations where these relationships are well established. And your population was mainly nurses so this can be compared but some caveats of course.

Response: We do not find it justified to get into a comparison with nurses in general, since working conditions differs thoroughly between working in a hospital and in primary care. We have clarified this at page 15-16, line 340-343.

Line 40, it is not clear what this means, our results differ..... If it is only suggested in other sectors, maybe this needs rewording so that it is clearer that other studies have identified lean as associated with increased MSP risk.

Response: We have adjusted the sentence and hope it is clearer now, page 16, line 345-347.

Line 48 this is very true, in relation to occupational context but this is why it is not clear why the results are surprising that the "people" variable was significant. We have added text that the result associated with "people" can be due to working conditions that can be risk factors. This is clarified at page 15, line 335-338.

Page 19 Line 22 is should be are.

Response: Thank you, this is now adjusted.

Line 35-37 needs rewriting for clarity, too many commas, and unclear

Response: Thank you, this is now adjusted. Page 16, line 364-365

Line 44 remove which even, and just say had a relatively low response rate.

Response: Thank you, this is now adjusted. Page 17, line 368-369.

Some mention of why you would not expect a change in MSP over one year would be useful, but also why you might expect a change in Lean measures, even if this was not done but it is a clear limitation, that is how sustainable were the Lean initiatives.

Response: At page 17, line 383-386 we have added text that suggests why we did not observe any changes in musculoskeletal complaints between T1 and T2. As for the change in Lean; we have, indeed, measured Lean at both T1 and T2. As an example, figure 2 illustrates the change in Lean maturity between these two points in time.

Some further explanation of what was done might actually help with interpretation of the results because if Lean is considered the intervention, then to what extent was the implementation successful. I realise this is not the intent of the article but nonetheless it does then provide some context as to what happened or did not happen and therefore the potential influence on the results. Response: We assessed Lean maturity twice with a one-year period in between; however not in a design assuming that (controlled) changes in Lean would occur, as in an intervention study. However, you are right that we did not collect data concerning what was done (or not done). We explain this as a limitation at page 17, line 378-380.

Minor comment for the start of the discussion:
 the use of the term prevalence's needs reviewing, used twice in the first three lines. Could be The musculoskeletal prevalence reporting the current study is higher than in the general or similar. Response: Thank you for helping us improving the readability. The sentence is now adjusted, page 14, line 315-316.

Reviewer: 2
 Dr. Victor Hoe, University of Malaya, Kuala Lumpur
 Overall, the manuscript has been well written, and it is easy to read and understand.
 Response: Thank you!

The clarity could be improved further by explaining in more detail the concept of Lean in Health care. Response: A clarification of Lean in healthcare is now given at page 4, line 86-90. After line 89 we further introduce the concept of Lean in healthcare.

It would be good also to provide a background on the history of Lean implementation in Sweden. Response: A short background is now given at page 4, line 86-90.

There is one major issue that I am not clear about. What is the association between Lean and MSD? There is no clear pathophysiological linkage between Lean and MSD, especially when we are focusing on localized MSD or MSD at a single site. The risk factors for localized MSDs are different from generalized MSD, i.e., multisite MSDs. Most of the time localized MSDs is the result of issues related to physical ergonomics issue, e.g., bad posture, repetitive movement, etc. If the authors are focusing on multisite MSD then the linkage between Lean and MSDs may be more apparent. Multisite MSDs are found to be associated be psychological, organizational and management styles. I would suggest if possible for the author to conduct an analysis looking at multisite pain to see if there are any associations. Response: Thank you for stressing this issue. We have rearranged and added text at page 5, line 99-102, and page 5 and 6, line 115-117 to make the link between Lean and MSD clearer for the reader. It is correct that we only included single-site complaints and not multisite. The reason for this is that single-site pain is viewed as a milder precursor to multisite pain and therefore single-site pain may be considered an indicator even for increased and multisite complaints/pain in the future.

VERSION 2 – REVIEW

REVIEWER	Oakman, Jodi La Trobe University, Public Health
REVIEW RETURNED	15-Sep-2022
GENERAL COMMENTS	Thank you for for the most part the authors have addressed the queries.

	I remain unconvinced that a name should be included as a keyword, the other words should enable the article to be found. The new next needs to be reread carefully as there are some grammatical errors that have been introduced, and need revision to ensure the meaning is clear. Briefly, Line 115-122, there are other risk factors for musculoskeletal complaints beyond the biomechanical ones listed, this needs to be attended to as there are a range of psychosocial hazards highly relevant to health care. Should be higher BMI not larger. Line 125 not panorama maybe wide variety Line 127 not different prevalence maybe differences between the two areas or something like that and 317 reword, not really clear if not from Sweden
--	--

VERSION 2 – AUTHOR RESPONSE

Reviewer: 1	
Dr. Jodi Oakman, La Trobe University	
I remain unconvinced that a name should be included as a keyword, the other words should enable the article to be found.	The name is now removed. Page 3 Line 49
The new text needs to be reread carefully as there are some grammatical errors that have been introduced, and need revision to ensure the meaning is clear.	The grammatical errors have now been adjusted as well as we have made sentences more readable, both in new text and throughout the manuscript. Page 4 Line 86-89 Page 5 Line 99-102 Page 5-6 Line 121-128

	Page 15 Line 336-339 Page 15-16 Line 341-343 Page 16 Line 345-346 Page 17 Line 383-385
Briefly, Line 115-122, there are other risk factors for musculoskeletal complaints beyond the biomechanical ones listed, this needs to be attended to as there are a range of psychosocial hazards highly relevant to health care. Should be higher BMI not larger.	Psychosocial risk factors have now been added. Page 5 Line 117-118 We have adjusted the sentence and specify only BMI. Page 5 Line 118
Line 125 not panorama maybe wide variety	We have adjusted the sentence and chose to remove "wide variety". Page 6 Line 125
Line 127 not different prevalence maybe differences between the two areas or something like that	This is now adjusted. Page 6 Line 126-127
Page 15 and line 317 reword, not really clear if not from Sweden	The sentence has now been made clearer. The wordings "if not from Sweden" is deleted. Instead, the text

	“Only one study was conducted in a Swedish setting”. Page 15 Line 320-321
Reviewer: 1 Competing interests of Reviewer: na